# During haptic communication, the central nervous system compensates distinctly for delay and noise

**Jonathan Eden** [1,2]* , **Ekaterina Ivanova**[1,3] , **Etienne Burdet**[1]

**1** Department of Bioengineering, Imperial College of Science, Technology and Medicine, London, United Kingdom, **2** Department of Mechanical Engineering, the University of Melbourne, Victoria, Australia, **3** School of Electronic Engineering and Computer Science, Queen Mary University of London, United Kingdom

☯ These authors contributed equally to this work.
* eden.j@unimelb.edu.au

During haptic communication, the central nervous
system compensates distinctly for delay and noise.
PLoS Comput Biol 20(11): e1012037. https://doi.
org/10.1371/journal.pcbi.1012037

UNITED STATES OF AMERICA

**Data Availability Statement:** The experiment data
and simulation code are available on Zenodo at link
https://doi.org/10.5281/zenodo.10811993.

**Funding:** This work was supported in part by the
EU H2020 grant ICT-871803 CONBOTS and by the

## Abstract

Physically connected humans have been shown to exploit the exchange of haptic forces
and tactile information to improve their performance in joint action tasks. As human interactions are increasingly mediated through robots and networks it is important to understand
the impact that network features such as lag and noise may have on human behaviour. In
this paper, we investigated interaction with a human-like robot controller that provides similar haptic communication behaviour as human-human interaction and examined the influence and compensation mechanisms for delay and noise on haptic communication. The
results of our experiments show that participants can perceive a difference between noise
and delay, and make use of compensation mechanisms to preserve performance in both
cases. However, while noise is compensated for by increasing co-contraction, delay compensation could not be explained by this strategy. Instead, computational modelling suggested that a distinct mechanism is used to compensate for the delay and yield an efficient
haptic communication.

## Author summary

Increasingly humans are making use of networks and robots to coordinate haptic interactions through teleoperation. However, with networks there comes delays and noise that
can change both the force that is transmitted and how we perceive that force. The haptic
communication involved in joint actions, such as when moving a piano or performing a
pair spin, has been shown to improve performance, but how does delay affect this behaviour? We tested how participants tracked a moving target with their right hand while connected to a human-like robotic partner, when perturbed by delay or noise. Through a
comparison between noise and delay perturbation, in experimental performance and in
simulation with a computational model, we found that participants could from small

UK EPSRC EP/R026092/1 FAIRSPACE program (all to EB). The funders had no role in study design, data collection and analysis, decision to publish, or preparation of the manuscript.

**Competing interests:** The authors have declared that no competing interests exist.

values of perturbation identify if the perturbation was from delay or noise and that they adopted different adaptation strategies in each case.

## Introduction

How do humans succeed in carrying out skilled motor tasks together, such as when moving a piano or when skaters perform a pair spin? The results of recent studies on joint tracking tasks [1–3] suggest that these collaborations are supported by the partners exchanging their motion plan via the haptic channel [4, 5]. Importantly, performance benefits arise between connected partners of different skills [1], where each partner must integrate information from their visual and haptic channels, and where there is evidence that connected partners can have better task learning than when in a solo configuration [6, 7]. The ability to coordinate incoming sensory information from different modalities and with different time lags has been proposed to be critical for the central nervous system (CNS) to make sense of interactions with the environment [8, 9]. Here, the response to temporal delays is also of practical importance due to the transmission delays present when partners are connected by robotic systems, e.g. teleoperation in space applications with one partner on Earth and the other on a space station [10], or during remote physical training [11]. However, despite the physiological importance of sensorimotor delays and its noted affect on some sensory modalities [12] and the perception of haptic interaction [13, 14], how it influences haptic communication is not yet known. Therefore, we designed an experiment to investigate how haptic communication is affected by temporal delays present over digital connection [15].

This paper examines what mechanism physically connected individuals use to collaborate despite delayed haptic feedback. One possibility is that the CNS ignores the delay (*no compensation strategy*, Fig 1A), in which case haptic communication (that can be modelled as in [4]) takes place as normal, such that performance should degrade with increasing delay. However, the CNS is known to inconspicuously fuse signals with different timing information [8], e.g., recognising the relationship between the visual signal of lightning and the delayed audio of thunder. This ability may extend to haptic communication, where sensor fusion occurs across partners [4]. Here, the evidence suggests that haptic communication arises due to the CNS identifying the interaction with the partner as task-relevant and optimally combining their own and the partner's sensory information by considering the respective noise of each information source [4]. Corresponding to these results, the mismatch of the delayed haptic information from the partner may instead be compensated as if it was an additional noise on the information derived from the partner (*compensation as noise strategy*, Fig 1B). Alternatively, the CNS may be able to identify that the haptic information has been delayed and actively compensate for it, using either a temporal [16] or state based [17] mechanism, thus enabling the extraction of more specific information from this signal (*compensation by delay prediction* strategy, Fig 1C).

To study how temporal delay affects haptic communication and to test these three possible approaches, we investigated how participants (connected to a human-like robot controller [5]) tracked a target moving along a multi-sine function with their dominant arm's wrist flexion/extension. The participants moved an individual handle of the Hi5 dual robotic interface [18] to track a target with a cursor on their own monitor (Fig 2A). They were physically connected by a virtual spring to the *robotic partner* of [4, 5], which has been shown to induce similar interaction perception, behaviour and learning to a human partner [6, 19]. This enabled the systematic investigation of the effect of temporal delay and noise in haptic communication. In

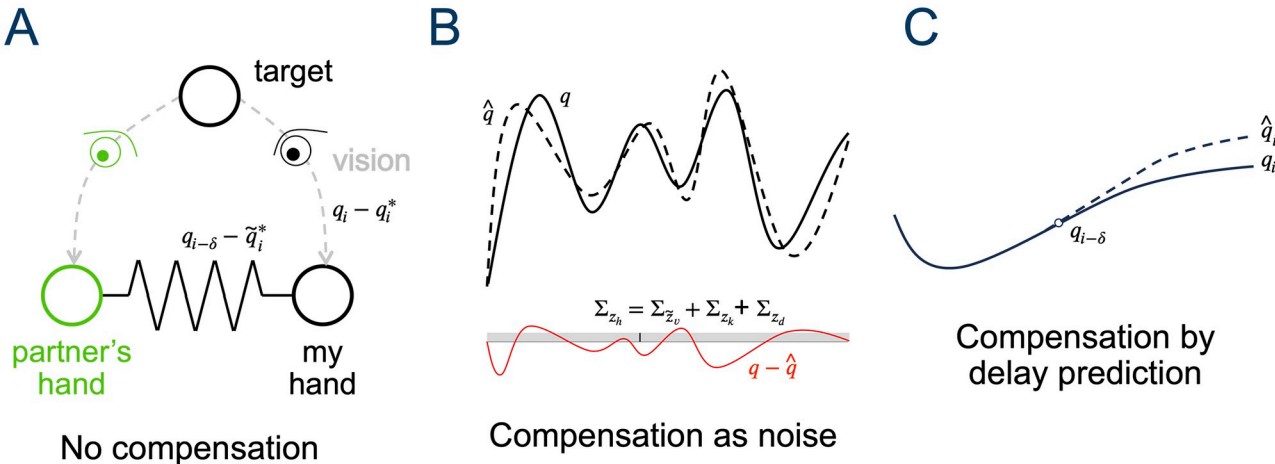

**Fig 1. Possible neural mechanisms to deal with a delayed interaction with a partner.** A: Haptic communication with *no compensation*, where the CNS understands that the haptic feedback is related to the visual task and uses it (without adjusting for delay) to infer the partner's motion plan that is then combined with their own motion plan. B: *Compensation as noise* mechanism in which the CNS, additionally to A, considers the delay as an additional noise to be filtered. C: *Compensation by delay prediction* mechanism in which the CNS, additionally to A, identifies the temporal delay and uses this knowledge to make a prediction (dashed line) of the partner's motion using the delayed haptic feedback. This prediction may deviate from the true trajectory if the partner model does not match their behaviour in the time after the delayed feedback.

this paper, we first reanalysed the 20 participant's data from [15] to investigate the effect of temporal delays up to 540 ms. To analyse whether the CNS compensates for delay as if it was noise (*compensate as noise*), we further tested the effect of noise in the haptic connection through a new group of another 20 participants. As muscle co-activation control is a known strategy to deal with noise [20, 21], the activity of a wrist flexor/extensor muscle pair was recorded. A questionnaire was used to compare perception of both types of perturbations. Our new experimental findings showed that while the noise data displayed a trend consistent with co-activation control, the delay data showed a different tendency. These findings were also validated against a computational model for haptic communication [5], where by extending the model we investigated the response to different possible mechanisms used by the CNS to compensate for temporal delays and reproduce the observed experimental findings.

## Experimental results

The experiment consisted of six blocks of target tracking (Fig 2B) conducted on the Hi5 robot with or without interaction with a *robot partner* (RP) (see Methods). Participants were split evenly into two groups of 20, where each group was defined by the perturbation added to the interaction: *delay*; or *noise*. For each group, the initial block was a *solo* condition in which the participants tracked the target without any connection. This was followed by five additional blocks in which a connection to the robot was provided and a perturbation on that connection was increased after each block. Delays of {0, 20, 60, 180, 540} ms were added to the haptic feedback in the *delay group* and random noise torques with standard deviation {0, 8, 22, 67, 184} mNm were added in the *noise group*.

### Perception of delay and noise

After each experimental block, we asked participants to provide information about their perception of the haptic interaction by answering a questionnaire (see Supporting information). In both the delay and noise groups, participants clearly identified the presence of forces in all

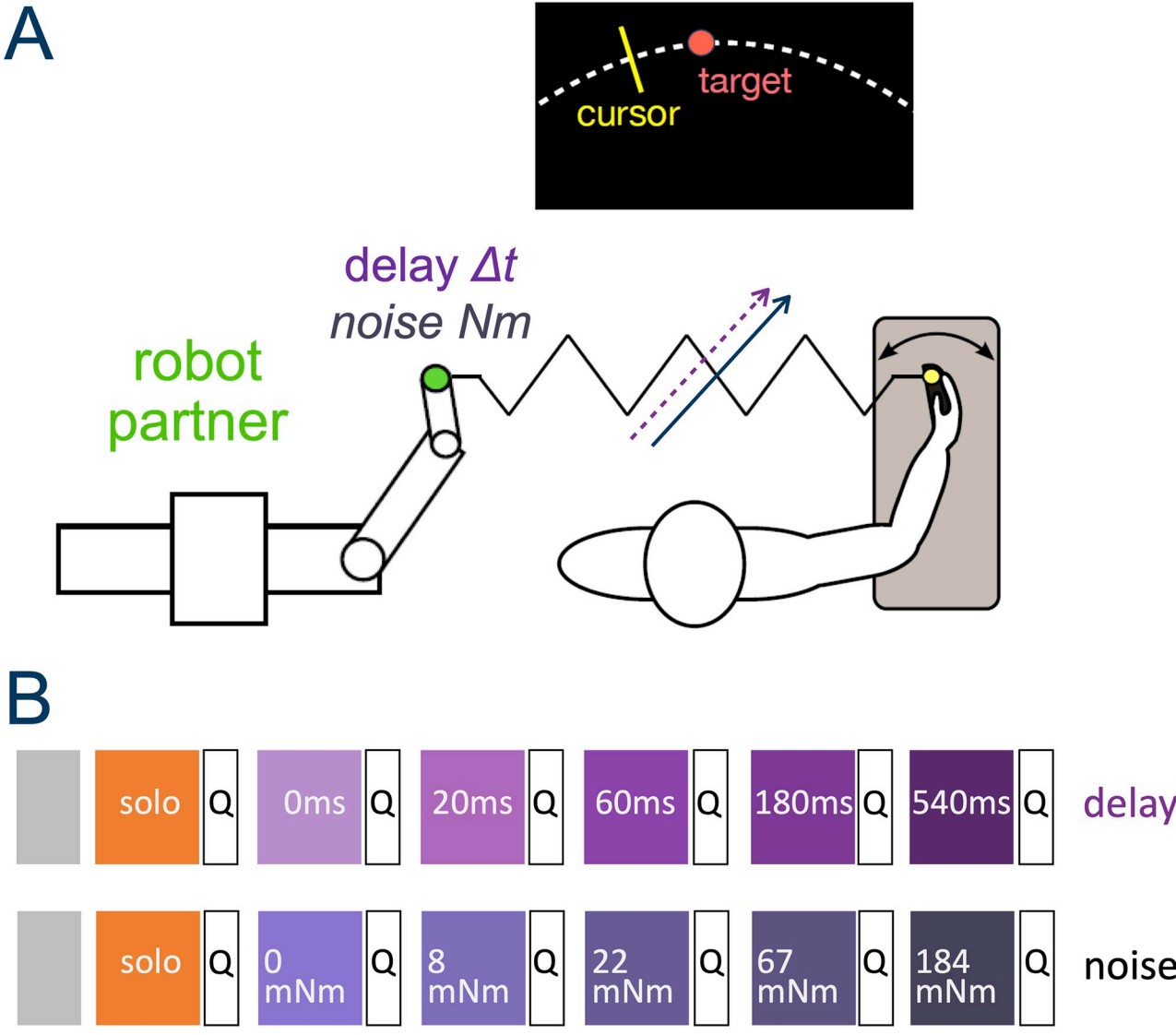

**Fig 2. Experiment description.** A: Participants tracked a randomly moving target with their wrist flexion/extension movement while being connected to a reactive robot partner (RP). B: The experimental protocol for the delay group [15] and noise group. The grey boxes represent the familiarization/washout trials and the colourful blocks are the experimental conditions. Both groups started with the solo condition (i.e. without interacting with a RP). They then were connected with a RP without delay/noise. Subsequently, the delay/noise was increased in every block. After each block, participants were asked to fill in a questionnaire. The order of the blocks was the same for all participants.

interaction blocks (Item 3: "During the task it seemed like I felt haptic forces", Friedman test, delay: $\chi^2(5) = 59.644$, $p < 0.0001$; noise: $\chi^2(5) = 57.847$, $p < 0.0001$), such that the solo condition was distinguished from all other conditions ($p < 0.05$ for all pairwise comparisons to solo condition with post-hoc Wilcoxon tests).

The questionnaire also asked participants if they perceived delay (Item 10) and noise (Item 6) during each interaction block through a 5-point Likert scale from "strongly disagree" to "strongly agree" (see Fig 3). In the delay group, participants appeared to clearly perceive delay (Item 10, Fig 3A) even at small applied delay values, where the perception was always different to that of the no-delay condition ($\chi^2(5) = 34.121$, $p < 0.0001$; $p < 0.05$ for all pairwise

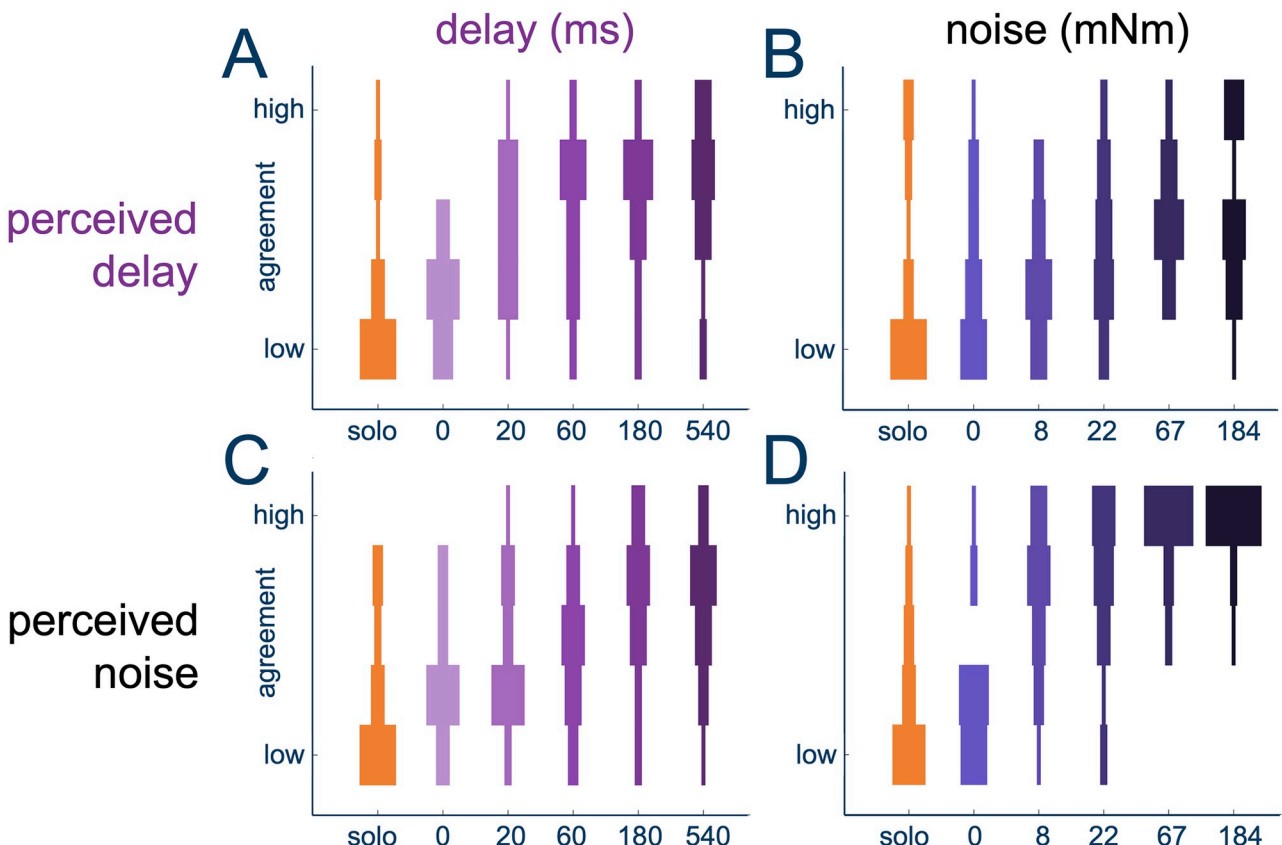

**Fig 3. Perception during the interaction with a RP perturbed by delay or noise.** (A) and (C) show the perception of delay and noise, respectively, for the delay group, while (B) and (D) show this perception for the noise group. Note the 5-point Likert scale goes from strongly disagree ('low') to strongly agree ('high').

comparisons between 0 ms and 20–540 ms delay groups). Moreover, participants disagreed that they perceived a delay in the 0 ms group ($p = 0.8281$ for the one-group comparison with the "disagree" value of Item 10). The delay group's perception of noise (Item 6, Fig 3C) also changed depending on the applied delay ($\chi^2(5) = 34.217$, $p < 0.0001$). However, in contrast to their delay perception, the participants only had a higher perception of noise (relative to the no-delay condition) at the two highest applied delay levels, 180 and 540 ms (both $p < 0.05$).

Participants in the noise group clearly perceived noise ($\chi^2(5) = 66.411$, $p < 0.0001$) as shown in Fig 3D. In contrast to the perception of noise in the delay group, noise group participants perceived the presence of noise from the smallest applied noise level ($p < 0.01$ for all pairwise comparisons between 0 mNm and 8–184 mNm noise conditions). In this group, the perception of the delay (Fig 3B) also changed with the applied noise level ($\chi^2(5) = 23.956$, $p = 0.00022$), however, none of the conditions were found to be clearly different from each other (all $p > 0.05$).

In summary, in both delay and noise groups, participants were able to recognise the presence of their respective perturbation from its smallest value. There was also some increase in perception of the non-adjusted factor in each group. However, there was limited confusion with a clear perception of the incorrect factor (compared to the solo condition) only reported in the delay group for its two highest delay levels. This indicates that the participants were able

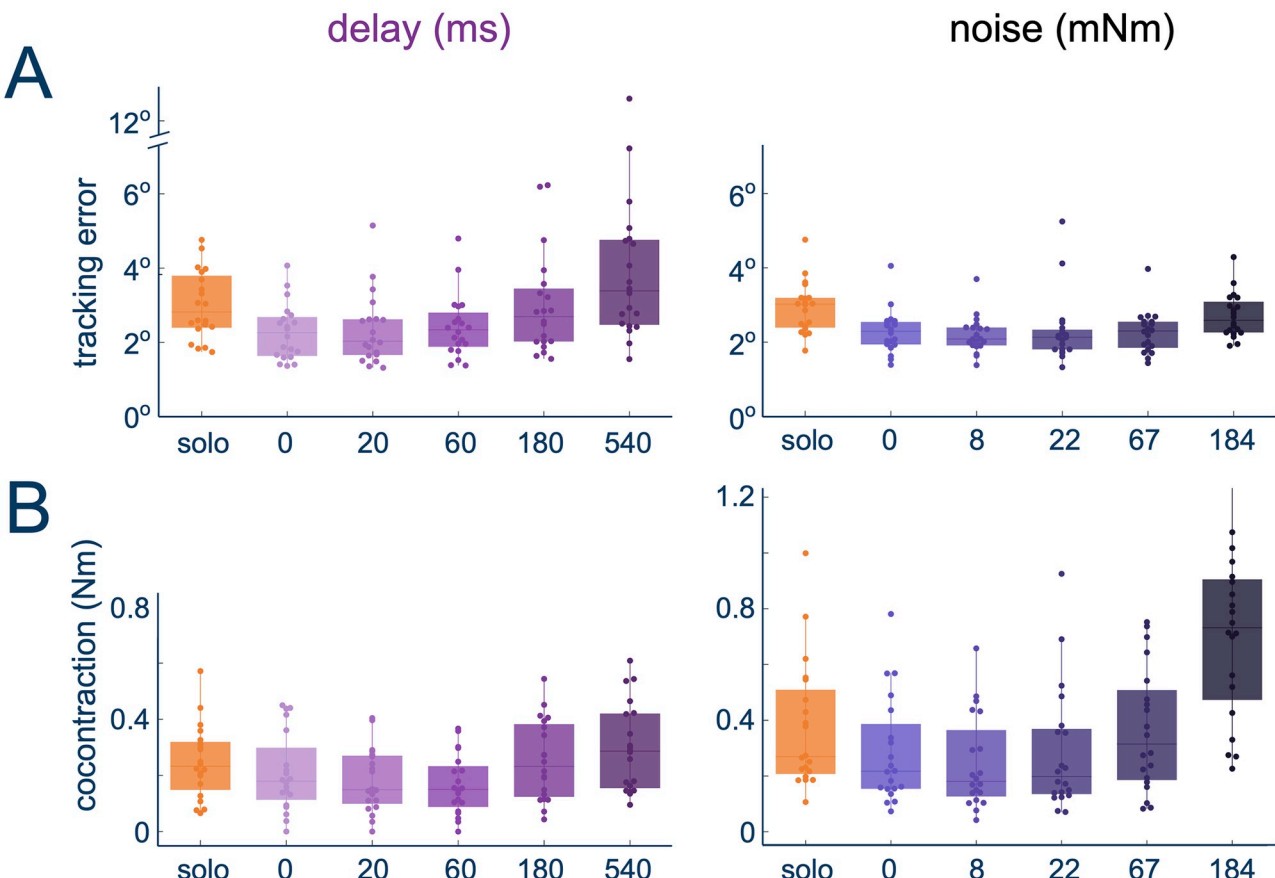

**Fig 4. Performance and effort during the interaction with the robotic partner with delay (left) or noise (right) perturbation: Tracking accuracy (A) and co-contraction (B).** Each dot represents the average value in each block for one participant.

to perceive and distinguish delay from noise even without any knowledge of the experiment perturbation types.

## Different behaviours induced by delay and noise

We then analysed the participant's performance (Fig 4A) to determine if the participants benefited from the haptic interaction (as has been previously observed in [4, 5]) and if they were able to compensate for the applied disturbances. Here, since they were instructed to track the target "as accurately as possible", performance was assessed through the root mean squared error relative to this target. In both groups, the robot partner condition affected the performance (delay: $\chi^2(5) = 63.143$, $p < 0.0001$; noise: $\chi^2(5) = 62.257$, $p < 0.0001$). Here, as in [4, 5], for both groups the addition of the robot partner (and the potential for haptic communication) improved the participant performance (solo vs. 0 for delay: $W = 210$, $Z = 3.9199$, $p < 0.0001$; and noise: $W = 210$, $Z = 3.9199$, $p < 0.0001$). However, while the participants were always able to compensate for the perturbation in the noise group such that the performance was always better to that of the solo condition (all $p < 0.04$) and not distinguishable from the condition working with a RP without perturbation ($p > 0.05$ for pairwise comparisons between 0-noise and 8, 22, 67 mNm), this was not the case for the delay group. Instead, here although the participants' performance was not clearly affected by small applied delays (0 vs. 20 ms: $W = 82$, $Z$

$= -0.85865$, $p < 0.4091$; 0 vs. 60 ms: $W = 61$, $Z = -1.6426$, $p = 0.3162$), they were not able to compensate for the larger applied delays (0 vs. 180 ms: $W = 23$, $Z = -3.0613$, $p = 0.0073$; 0 vs. 540 ms: $W = 3$, $Z = -3.8079$, $p < 0.0001$) such that at 540 ms their performance was worse than that of the solo condition (solo vs. 540 ms: $W = 31$, $Z = -2.7626$, $p = 0.0211$). This shows that while both groups improved from the addition of haptic communication, participants were only able to compensate for small delay values while they could compensate for all tested noise disturbances.

To understand if participants were using the previously observed co-activation noise compensation strategy [22] to deal with both types of applied perturbation, we analysed their co-contraction (Fig 4B) computed as the minimum measured absolute muscle activity of the flexor and extensor muscles over a trial (see Methods). Participants changed their co-contraction in response to both applied perturbation types (delay: $\chi^2(5) = 40.829$, $p < 0.0001$; noise: $\chi^2(5) = 58.6$, $p < 0.0001$). However, while the noise group displayed the previously observed noise compensation trend of increasing the median co-contraction with each subsequent applied perturbation level after the 8 mNm condition such that the co-contraction at the highest applied noise level was clearly larger than the no delay co-contraction (0–184 mNm: $W = 0$, $Z = -3.9199$, $p < 0.0001$), this was different in the delay group. Instead here, the participants held their co-contraction levels roughly constant for all small applied delay values ($p > 0.05$ for all pairwise comparisons between solo, 0 ms, 20 ms and 60 ms conditions except for solo-60 ms: $W = 25$, $Z = -2.9866$, $p = 0.0169$, where the co-contraction was slightly lower). There was then an increase for the two highest applied delay values (60 ms-180 ms: $W = 1$, $Z = -3.8826$, $p < 0.0001$; 60 ms-540 ms: $W = 0$, $Z = -3.9199$, $p < 0.0001$), in which the co-contraction level was not clearly different between the two levels ($W = 25$, $Z = -2.9866$, $p = 0.0169$). While the increase of co-contraction in the noise group was not concurrent with an increase in tracking error, as would be predicted if using the noise compensation strategy, the observed increase in co-contraction for the delay group coincided with an increase in tracking error.

Finally, we examined if the two different perturbations affected the user trajectories, through the cross-correlation delay between the reference and user trajectory (Fig 5A), and the motion smoothness (Fig 5B) as computed by the SPARC measure [23]. Here, the cross-correlation delay measured the difference between the target and participant trajectory, such that if the delayed haptic information of the delay perturbation was not compensated for, it would have resulted in increased cross-correlation delay. Moreover, the smoothness measured if the participant's motion had greater frequency spectrum complexity as would occur for an uncompensated noise perturbation.

The addition of the robot partner and perturbations affected the cross-correlation delay of both groups (delay: $\chi^2(5) = 36.564$, $p < 0.0001$; noise: $\chi^2(5) = 48.724$, $p < 0.0001$), where interestingly working with the robot partner without perturbation resulted in a reduced cross-correlation delay (both $p < 0.0001$), suggesting that the haptic connection may have aided a quicker reaction. For the delay group, the cross-correlation delay in high-delay conditions was higher compared to smaller delay conditions (20—540 ms: $W = 20$, $Z = -3.1733$, $p = 0.0007$; 60—540 ms: $W = 11$, $Z = -3.5096$, $p = 0.0005$; 60—180 ms: $W = 5.5$, $Z = -3.6028$, $p = 0.0003$) suggesting that at those delay conditions participants struggled to counteract the delay. In contrast, for the noise group, the cross-correlation delay was higher in the solo and 0-noise condition ($p < 0.0001$ for all pairwise comparisons with solo; 0–8 mNm: $W = 147$, $Z = 2.6787$, $p = 0.0437$; 0–22 mNm: $W = 148$, $Z = 2.7222$, $p = 0.0366$; 0–67 mNm: $W = 163$, $Z = 3.3752$, $p = 0.0019$) and all other conditions were similar to each other ($p > 0.05$ for all other pairwise comparisons). This suggests that the presence of noise may have made the participants more reactive.

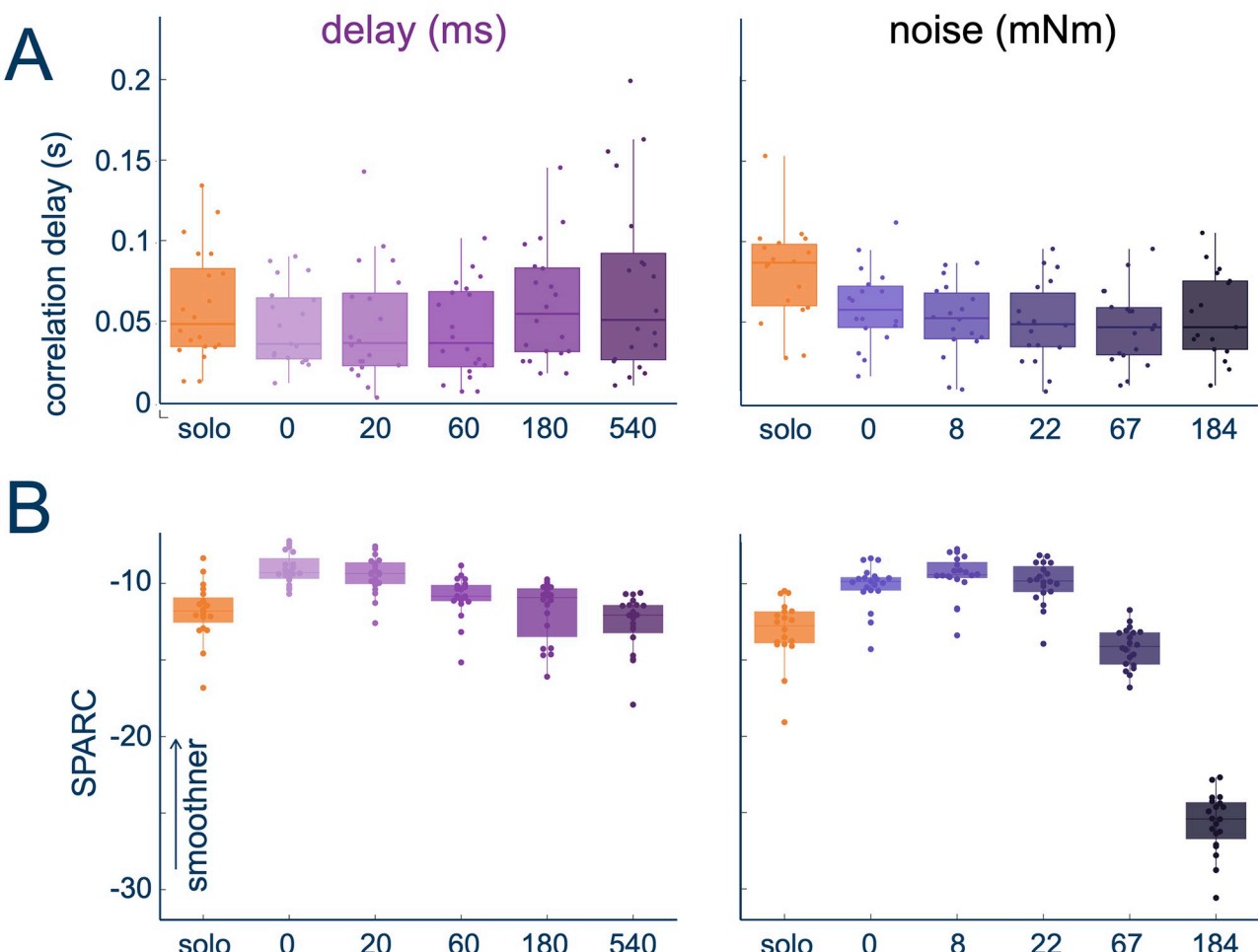

**Fig 5. Lag and noisiness during the interaction with the robotic partner with delay (left) or noise (right) perturbation: Cross-correlation delay (A) and smoothness (B).** Each dot represents the average value in each block for one participant.

The participants smoothness was also affected by the perturbation (delay: $\chi^2(5) = 75.771$, $p < 0.0001$; noise: $\chi^2(5) = 93.486$, $p < 0.0001$), where as has been previously observed [19], the addition of the robot partner without perturbation resulted in smoother motion (both $p < 0.0001$). In both cases, the high perturbation levels were different to that of the no perturbation robot partner (delay: $p < 0.05$ for 0–180 ms and 0–540 ms; noise: $p < 0.05$ for 0–67 mNm and 0–184 mNm) suggesting that both perturbations resulted in a more jerky motion.

These results confirm that haptic communication improves interaction performance. While the observed behaviour is then affected by both the delay and noise perturbations, the effect is distinct in the two groups. For the noise group, participants appear to compensate as would be predicted by the co-activation compensation strategy, where their co-contraction increases with the applied perturbation level, and while there is an observed degradation of motion smoothness, there is no tracking performance that is worse than that of the solo condition. In contrast for the delay group, while there is an increase in co-contraction for the two highest applied perturbation levels, this increase does not result in compensation for the performance. Instead, participants appear to use a distinct method to counteract for the applied delay where only at the highest levels of injected delay was there a different lag to the other

delay perturbation levels, and this was still not different from their own lag in the solo condition.

## Simulation results

The experimental results suggest that the response of the noise group is consistent with that predicted by the co-activation compensation strategy. However, the delay group behaviour was not consistent with this predicted behaviour, since the co-contraction did not consistently increase with a larger applied delay level. Could the results be explained strictly from the interaction mechanics, without any compensation? To evaluate this, we simulated the experimental scenario (with the same number of trials and blocks) of a participant being connected (by a virtual spring) to a partner with delayed haptic feedback. Here both partners were simulated using the model of haptic communication presented in [4, 5], which is based on four principles: i) The CNS of each participant is able to identify that the haptic feedback that they are receiving is related to the (visual) tracking task; ii) By using a model of tracking control, the CNS can extract from the haptic feedback an estimate of the partner's tracking error; iii) The participants then each combine their own and partner motion information in a stochastic optimal way, yielding a Bayesian sensor fusion of visual (own) and haptic (partner) information; iv) The viscoelasticity of the haptic connection to the partner is incorporated into the sensor fusion, where more compliant connections are considered as having additional uncertainty. To simulate the effect of the delayed haptic feedback on the haptic communication model of [5], we created 20 simulated participants, where for each measured tracking error (skill level) of an experimental participant there was an equivalent simulated participant. For each simulated participant, we delayed the virtual spring torque for one of the participants with a delay value matching those of the experimental condition.

When both partners were modelled as in [5] such that there was *no compensation*, the performance (Fig 6C) did not match that observed in the experimental data (Fig 6A) for participants with lower skill. Here, while there was little difference for small applied delay values (and participants with higher skill), the performance for lower skilled participants could become unstable at larger delay levels such that it overshot the participant performance. Therefore, from the differing behaviour of low skill simulated participants, we conclude that the participants had a mechanism to compensate for the delay.

Participants did appear able to recognise delay (Fig 3A). However, they may have adopted different strategies for its compensation which could lead to the observed motion characteristics. To investigate this compensation mechanism we extended the model developed in [5] to consider one possible *compensation by delay prediction* model as well as *compensation as noise* (see Methods). In the *compensation as noise* simulation, this consisted of modifying iv) such that both the viscoelasticity of the haptic connection and the delay of the feedback of that connection were considered as additional sources of uncertainty for the haptic information source (Fig 1B). In the *compensation by delay prediction* simulation, it was instead assumed that as a part of i) and ii) the participant would be able to identify the delay and use their model of the interaction and delay to predict the future value for the haptic feedback (Fig 1C).

The simulation results exhibit differences between the two models. Here, the *compensation as noise* simulation (Fig 6E) incorrectly predicted the delay response to have a similar trend to what was observed for noise group (Fig 6B), with a roughly constant tracking error behaviour across the applied delay levels. In contrast, the *compensation as delay prediction* simulation (Fig 6D) predicted an increase in tracking error and in the variance across participants for the higher delay levels.

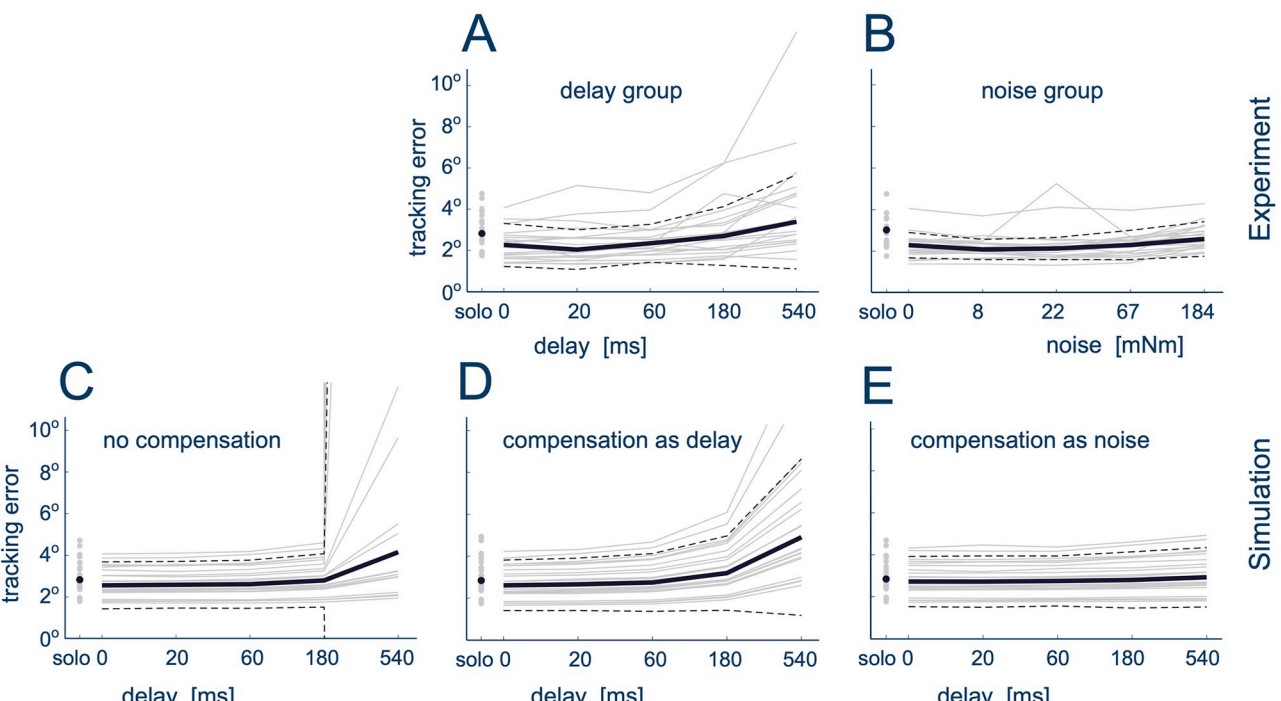

**Fig 6. Participant specific (and average) tracking performance for the experimental data and in the simulation to analyse the mechanism to compensate for temporal delays.** Experimental results are shown for the delay group (A), as well as for the noise group (B). All simulated results consider only delay compensation, where results are shown for: no compensation (C), compensation as noise (D) and compensation by delay prediction (E). In all subfigures, the grey lines indicate individual participant (real or simulated) performance while the thick solid black line and the dashed black lines denote the median performance and interquartile range of performance across all participants.

To understand the differences in the proposed models, we compared the variance between the experimental data and the simulated data for noise, delay and no-adaptation compensation. The F-test did not show a clear difference in variance between the experimental data and the compensation as delay prediction simulation for any level of delay (all $p > 0.1$). In contrast, for the highest level of delay (540 ms), the results showed a clear difference between the experimental data and both the no-compensation simulation and the compensation by noise simulation (no-compensation: $F_{(19, 19)} = 10320643$; $p < 0.0001$; noise: $F_{(19, 19)} = 0.1617$; $p = 0.0002$) and a tendency for different variance at the 180 ms level (noise: $F_{(19, 19)} = 0.4244$; $p = 0.0692$; no-compensation: $F_{(19, 19)} = 0.4044$; $p = 0.0554$). No difference for other levels of delay was found for these strategies (all $p > 0.1$).

## Discussion

This paper investigated the mechanism of haptic communication during a tracking task by considering the response to delay and noise perturbations. Our results indicate that participants can still exploit haptic communication in the presence of both small delays and moderate levels of noise, where haptic communication resulted in improved smoothness, accuracy and smaller correlation delays. Interestingly, the participants were able to correctly recognise the presence of delays and noise from their smallest values with limited confusion between the correct perturbation and other possible perturbing modalities. They then appear to compensate for each of these two different perturbation types with different strategies.

While our findings showed similar tracking error and smoothness improvements resulting from haptic communication as had been previously observed [1, 4], we further observed a reduction in the cross-correlation delay that had not been previously investigated. This indicates that participants reacted faster to the moving target during human-like robot interaction. This may be explained by two factors: i) since the robot partner is also tracking the target, the interaction force assists the user such that it speeds up their response; and ii) the user uses haptic communication to more quickly update their motion plan. While it is difficult to separate these two factors, it is noted that the observed improvement in cross-correlation delay was still present with all noise levels and with the small applied delay values. Both of these perturbations would have meant that the interaction force sometimes hindered target tracking. The reduced lag may therefore suggest that the information exchange of haptic communication not only improves the quality of the estimation but also further aids participants to react quicker.

In response to increased noise, our experimental findings indicate that while participants had a clear reduction of smoothness, their co-contraction showed an increasing trend with increasing applied noise. This result is consistent with the predictions made by the co-activation compensation strategy [20, 22], therefore indicating that humans use this same strategy during haptic communication. Interestingly, the cross-correlation delay reduced after the application of the noise torque. This finding, that the application of noise can make participants react quicker, merits further investigation as it would not be predicted by existing haptic communication models [4, 5] and may suggest an additional mechanism within haptic communication.

The delay group showed different behaviour to the noise group, where the performance became worse for high delay values, and at these values there was an observed decrease in smoothness and increase in cross-correlation delay and co-contraction. In the context of haptic communication, these results could be obtained through distinct response mechanisms such as: i) participants do not compensate for delay. Here, the lack of a clear performance changes for small delay values is reflective of the delay having a minimal effect on the system dynamics at these values; ii) participants do not explicitly compensate for delay and instead identify the delay as an additional source of uncertainty. For large delay values, performance degrades due to the participants incorrectly modelling the delay as noise; and iii) participants possess a distinct compensation mechanism for delay. Here, the compensation mechanism may be imperfect leading to the observed changes in behaviour.

Our simulations indicate that while the *no compensation* model can result in similar performance for some participant skill levels, it is not robust, such that for the extreme cases (corresponding to the lower skill level experimental participants) it would predict unstable performance (Fig 6C). The lack of such cases in our experimental results (Fig 6A) suggests the presence of a compensation mechanism for delay, as has been observed in multi-sensory integration [24] and in the adaptation to applied force [16, 25, 26] and visual [17, 27] delays. Here, it is also noted that participants were able to perceive delay even for small values (Fig 3A). While, there is evidence that the mechanisms for perceiving delay and adapting to it are not necessarily the same [16], this has suggested that compensation occurs without perception but not the other way.

The simulation results also suggest that the delay group participants did not increase their co-contraction in a manner consistent with the compensation by noise compensation strategy. Our findings therefore appear to be consistent with a strategy of *compensation by delay prediction*, where the CNS identifies the delay and distinctively compensates for it. Our simulations consider this compensation to be an explicit time-based compensation as has been observed for the multisensory integration of information across the visual and haptic channels [24] and in other tracking tasks with delay [26, 28]. However, it is worth noting that other studies have

shown that force delay may instead be compensated for using a mixture of current and delayed state information [25], or that visual delay may be compensated for using state-based mechanisms such as assuming altered impedance characteristics [27] or scaling of the measurement [17] properties.

Given that previous studies have shown that the perception of delay is associated to a perception of stiffness [13, 14] and that increasing compliance can be modelled by an increase in uncertainty in Bayesian integration [5], the haptic communication model [4] could be updated to consider alternate delay compensation models through an updated haptic channel measurement with additional sources of uncertainty. Our current experimental results are however not able to distinguish between these different possible delay compensation strategies. Moreover, it is worth noting that the two applied delay levels for which noise was perceived (delays of 180 and 540 ms) both also coincided with an increase in co-contraction. It is therefore possible that participants used a mix of the *compensation by delay prediction* and *compensate by noise* strategies, where in response to their insufficient compensation participants identify noise and then try to compensate (ineffectively) through co-activation.

In summary, physical connections over the haptic channel can improve user performance and is robust to small to medium sized perturbations in the form of delays and noise. Our findings suggest that this compensation is made possible by the participant being able to uniquely perceive the presence of delay or noise and then compensate specifically to these different perturbations.

## Methods

### Ethics statement

The experiment was approved by the Research Ethics Committee of Imperial College London. Each participant gave informed formal written consent, and filled in a demographic questionnaire as well as the Edinburgh handedness form [29] before starting the experiment.

### Participants

The experiment was carried out by 40 participants (21 female, 19 male) without known sensorimotor impairment aged 24.02±3.19 years old. All participants performed the task with their dominant hand, where four participants were left-handed. Participants were divided into two groups that each experienced only one type of perturbation: human-robot interaction with i) time delay (data collected in [15]) or ii) with haptic (torque) noise.

### Experimental setup and protocol

The experimental task was designed to replicate the tasks of existing studies of haptic communication [19]. Two participants completed the experiment at the same time, with their dominant arm attached to the Hi5 dual robotic interface [18]. The participants were instructed that within the experiment they might experience a haptic interaction at their wrist. They were then visually separated from one another by a curtain and each participant's wrist flexion/extension movement interacted with the Hi5 interface, which was controlled at 1000 Hz, while the wrist angle data was recorded at 100 Hz.

Each participant was asked to track a moving target "as accurately as possible" using the wrist flexion/extension of their dominant hand (Fig 2A). The target trajectory was given by

$$q^{*}(t) \equiv 18.5^{\circ} \sin[2.031\,(t + t_0)]\,\sin[1.093\,(t + t_0)]\,, \quad 0 \le t \le 30\,s\,, \tag{1}$$

where to minimise the learning of the trajectory each trial started from a randomly selected

starting time $\{t_0 \in [0, 30] \text{ s} \mid q^*(t_0) \equiv 0\}$. The participant's wrist flexion/extension was connected to a *robotic partner* (RP) with angle $q_r$ through a virtual viscoelastic band described (in Nm) by

$$\tau(t) = 1.72\left[q_r(t_\delta) - q(t_\delta)\right] + 0.0286\left[\dot{q}_r(t_\delta) - \dot{q}(t_\delta)\right], \quad t_\delta \equiv t - \delta. \tag{2}$$

The robotic partner (RP) is a reactive controller [4, 19] that mimics human interaction behaviour. This includes accounting for the different skill levels of human participants, where the robot's skill level (as measured by the RMS tracking error) can be set. Here, the RP replicates human haptic communication behaviour through a sensory augmentation approach in which the information coming from the haptic connection is used to infer the partner's motion information [4]. This is then combined with their own motion information in a stochastically optimal manner, where less skilled agents have larger uncertainty in their measurements.

For Eq (2) the damping and stiffness constants were chosen to match the conditions of medium stiffness in [4], with which an interaction with an interactive agent was clearly perceived by participants [19]. In the *delay group* $\delta \in \{0, 20, 60, 180, 540\}$ ms was used for the delayed interaction torque (while the robot partner received the torque without delay). For the *noise group* $\delta = 0$, while the torque was perturbed by Gaussian noise $v$:

$$\tau_v(t) = \tau(t) + v, \quad v \in N(0, \Sigma_\eta), \tag{3}$$

where the standard deviation $\Sigma_\eta \in \{7.5, 22.5, 66.7, 184\}$ mNm was used.

Surface electrodes were used to record electromyographical (EMG) activity from the wrist flexor carpi radialis (FCR) and extensor carpi radialis longus (ECRL) muscles. This was calibrated through a process in which participants were asked to flex/extend while their wrist was locked by the device at 0° corresponding to the participant's most comfortable position. Each participant was asked to produce flexion and extension torques of $\{1, 2, 3, 4\}$ Nm for 2 seconds, first flexion then extension, with a rest period of 5 seconds between each activation to prevent fatigue. This EMG data was linearly regressed with the measured torque to estimate the relationship between muscular activity and torque. Then the *co-contraction* was computed as

$$u(t) \equiv \min\{\tau_f(t), |\tau_e(t)|\}, \tag{4}$$

where $\tau_f(t) \geq 0$ and $\tau_e(t) \leq 0$ are the flexor and extensor torques, computed from the respective EMG signals. The average co-contraction over all participants was computed from each participant's normalised co-contraction, calculated as

$$u_n \equiv \frac{\bar{u} - \bar{u}_{min}}{\bar{u}_{max} - \bar{u}_{min}}, \quad \bar{u} \equiv \frac{1}{T}\int_0^T u(t)\,dt, \quad T = 30\,s \tag{5}$$

with $\bar{u}_{min}$ and $\bar{u}_{max}$ the minimum and maximum of the means of all trials of the specific participant.

The experiment protocol is described in Fig 2B. In the initial solo block, each participant attempted five trials of the task without a haptic connection to be familiarised with the task and to minimise subsequent learning effects. In the main experiment, participants carried out six blocks, each of ten trials. Each block included seven experimental trials followed by three washout trials of the solo condition. The first experimental condition was without any interaction and in the following five blocks assistance from a robot partner was introduced for both delay and noise experiments. The robot partner's uncertainty was set after the solo block using a mapping that converted RMS error into measurement uncertainty [4]. Here the robot's skill was set to be equal to the deviation observed in the participant's tracking movement during

the final solo trial to ensure that the participant and the RP had similar skill level. This was chosen to ensure that there was no clear difference in the participant and RP's performance during the tracking trials. Each trial took 30 s and was followed by a 5 s break.

The delay and noise within the robotic assistance trials were increased from each block to the next with values {0, 20, 60, 180, 540} ms for delay and {0, 8, 22, 67, 184} mNm for noise. The sequence of the blocks with increasing level of perturbation was identical for each participant within delay and noise experiments. The delay levels were chosen to include small delay values considered in [30] and values greater than the threshold for performance loss found in [31]. The Gaussian noise values were then set to approximate the effect of the delay. Here, the noise torque standard deviation $\Sigma_\eta$ was set so that three standard deviations was equivalent to the likely maximum error torque caused by a given delay, which was approximated by $1.72 \cdot \max\{q^*(t) - q^*(t - \delta)\}$. After each block, the participants had to answer questions about their perception of the interaction (see Supporting Information S1 Text).

## Simulation framework

We evaluated the mechanism for delay via 20 simulated participants that were programmed with the computational model developed in [5] which matched the algorithm used by the RP. In this discrete time model, the control of the wrist is modelled as a linear controller that acts on a double integrator system which describes the wrist angle $q$. The state space dynamics at time-step $i$ are given by

$$\mathbf{q}_{i+1} = \mathbf{A}\,\mathbf{q}_i + \mathbf{B}(u_i + \tau_i), \quad \mathbf{q}_i \equiv \begin{bmatrix} q_i \\ \dot{q}_i \end{bmatrix}, \quad \mathbf{A} = \begin{bmatrix} 1 & dt \\ 0 & 1 \end{bmatrix}, \quad \mathbf{B} = \begin{bmatrix} 0 \\ dt/I \end{bmatrix}, \quad (6)$$

where $dt$ is the time differential and $I = 0.002$ kg m$^2$ the wrist's moment of inertia (as defined for experiments with the Hi5 robot in [5]). The control input $u_i$ is determined by the linear feedback control law

$$u = -L_p(q - \widehat{q^*}) + L_v(\dot{q} - \widehat{\dot{q}^*}), \quad (7)$$

where $\widehat{q^*}_i$ denotes the participant's estimate of the trajectory. $L_p$ and $L_v$ are the proportional and derivative gains determined to minimise a quadratic cost function of error and effort [20]. Moreover, the system is influenced by the haptic interaction torque $\tau_i$. This torque is set to 0 during solo trials. It instead acts as a spring and damper torque and is set as in Eq (2) for interaction trials connecting the agent to their partner whose dynamics similarly evolves with a form given by Eq (6).

The key feature of the model is that the partners improve their estimate $\widehat{q^*}$ through visual feedback and haptic information from the interaction. Here, their own target information is combined with the partner's target information $\widehat{\tilde{q}^*}_i$, as determined from the interaction force. This integration of the partner's target information is carried out through a Kalman filter in which the measurement $\mathbf{z}_i$ is given by

$$\mathbf{z}_i = \begin{bmatrix} q_i - q_i^* \\ q_i - \widehat{\tilde{q}^*}_i \end{bmatrix} + \begin{bmatrix} \mu \\ \tilde{\mu} \end{bmatrix}, \quad (8)$$

where $\mu \in N(0, \Sigma_{z_v})$ and $\tilde{\mu} \in N(0, \Sigma_{z_h})$. This assumes that $\Sigma_{z_v}$ characterises the visual noise naturally present in the participant's tracking, while $\Sigma_{z_h}$ characterises the haptic noise and is composed of the partner's visual noise $\Sigma_{\tilde{z}_v}$ and additional noise resulting from the virtual band

viscoelasticity $\Sigma_{z_k}$ such that $\Sigma_{z_h} = \Sigma_{\tilde{z}_v} + \Sigma_{z_k}$. In this way, the haptic interaction acts both as an input force that could help in target tracking and as a second sensor measurement that can improve the participant's understanding of their current task state.

**Delay compensation strategies.** Three delay compensation strategies are considered within the simulation to explore the features of the participants response to delay: i) no compensation; ii) compensation as noise; iii) compensation as delay prediction. In the *no compensation* strategy participants were simulated to directly use the above interaction model with a delayed estimation of the partner's error, i.e.

$$\mathbf{z}_i = \begin{bmatrix} q_i - q_i^* \\ q_{i-\delta} - \widehat{\tilde{q}^*}_i \end{bmatrix} + \begin{bmatrix} \mu \\ \tilde{\mu} \end{bmatrix}. \tag{9}$$

In the *compensation as noise* strategy participants were modelled to use the same delayed measurement as in Eq (9). However, they were assumed to consider the haptic signal to be noisier with an additional independent noise $\Sigma_{z_d}$ associated to the given delay level such that

$$\Sigma_{z_h} = \Sigma_{\tilde{z}_v} + \Sigma_{z_k} + \Sigma_{z_d}. \tag{10}$$

The magnitude of this additional delay generated noise was determined by simulating the participant's solo performance with the delay and then mapping the resulting error to a noise value through the error to noise regression determined in [5].

Finally, in the *compensation as delay prediction* strategy the participants were instead modelled to be able to identify both the presence of the delay as well as its magnitude. They were then assumed subsequently adjust their haptic measurement through forward integration with the known system dynamics Eq (6) and visual target sequence. Here the current error estimate $q_{i-\delta} - \widehat{\tilde{q}^*}_i$ was updated by iteratively applying Eq (6) (without the unmeasured torque $\tau_i$) for the number of delayed time steps (given as $\delta/dt$).

## Data analysis

To investigate the delayed force exchange's effect on participant performance, the Root-Mean-Square Error (RMSE), the smoothness metrics SPARC [23], the cross-correlation delay and the co-contraction were analysed. The *cross-correlation* delay corresponds to the time interval between the target's movement and the participant's resulting motion and was calculated as the time lag at which the cross-correlation between the target and participant's positions was the highest. To understand how participants perceived the changes in delay, a questionnaire composed of a 5-point Likert-scale item was analysed (see the question list in Supporting information). To check the consistency of the questionnaire responses for Items 2–12, composite reliability and Cronbach's alpha were used to determine whether responses were consistent between items. This gave values of 0.86 and 0.87, respectively, which confirmed the consistency between the item responses, and indicates that the items were not redundant.

Since we could not directly compare the delay and noise magnitudes, a separate analysis was conducted for each perturbation group, and the group tendencies were then qualitatively compared. Since each metric was found to not be normally distributed, the influence of the perturbation on each metric was explored through Friedman tests. Post-hoc analysis between individual perturbation levels was conducted using a paired Wilcoxon sign-rank test with the Hommel adjustment to control the family-wise error rate. For each objective value (RMSE, SPARC, cross-correlation delay, co-contraction) the analysis was conducted for each participant using the averaged value over all trials in a block. To compare the variance between the

simulated data of for the compensation by noise, compensation as delay prediction and no compensation models and the experimental data, an F-test was used for each delay level.

## Supporting information

**S1 Text. Questionnaire items.**
(DOCX)

## Author Contributions

**Conceptualization:** Jonathan Eden, Ekaterina Ivanova, Etienne Burdet.

**Formal analysis:** Jonathan Eden, Ekaterina Ivanova.

**Funding acquisition:** Etienne Burdet.

**Investigation:** Ekaterina Ivanova.

**Methodology:** Jonathan Eden, Ekaterina Ivanova.

**Software:** Jonathan Eden.

**Visualization:** Jonathan Eden, Ekaterina Ivanova, Etienne Burdet.

**Writing – original draft:** Jonathan Eden, Ekaterina Ivanova, Etienne Burdet.

**Writing – review & editing:** Jonathan Eden, Ekaterina Ivanova, Etienne Burdet.

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
