## [Decision Letter · Decision Letter 0]

11 Jul 2024

Dear Dr Eden,

Thank you very much for submitting your manuscript "During haptic communication, the central nervous system compensates distinctly for delay and noise" for consideration at PLOS Computational Biology.

As with all papers reviewed by the journal, your manuscript was reviewed by members of the editorial board and by several independent reviewers. The reviewers were generally positive about the paper, but did raise some substantial concerns about the modeling for the delay condition. There is a substantial literature on adaptation to delayed feedback which has has argued against the simple delay prediction model put forward here. This literature is largely overlooked, however. In light of the reviews (below this email), we would like to invite the resubmission of a significantly-revised version that takes into account the reviewers' comments and in particular addresses the issues raised by the reviewers surrounding the modeling of behavior in the delay condition.

We cannot make any decision about publication until we have seen the revised manuscript and your response to the reviewers' comments. Your revised manuscript is also likely to be sent to reviewers for further evaluation.

Sincerely,

Adrian M Haith

Academic Editor

PLOS Computational Biology

Andrea E. Martin

Section Editor

PLOS Computational Biology

Reviewer's Responses to Questions

**Comments to the Authors:**

Reviewer #1: The authors present an interesting study of how human actors might account for noise and delay when interacting haptically with humans or robotic assistants to perform joint action tasks. I find the experiments and analysis to be well done and modelling of human behaviour to be convincing and innovative.

I have one methodological question as to how to differentiate between mechanical effects of the haptic interaction on performance versus a Bayesian interpretation of the interaction in terms of optimal control. And I have some suggestions on how the author's might interpret their results with respect to both haptic perception in the face of delays and in terms of a potential link to the classic binding problem.

Lines 141 and 154: It is indeed interesting that adding the RP without perturbation both increased the smoothness and decreased the correlation delay. Indeed, I have a question about the comparison between solo and with RP but without added noise or delay. This may have been covered in previous publications, but I have not read them recently and in the interest of time (my review is late) I will pose my questions here. At line 276 it is stated " the information coming from the haptic connection is used to infer the partner’s target, which is then combined with their own target in a stochastically optimal manner." In the experiments reported here, what is the RP's "own target"? Was it an accurate version of the actual target? This would mean that the RP is a priori assisting the human subject and the human can infer from instructions or deduce that the haptic information should be helpful and use that information to modulate their own muscle activations to control the wrist. Or is the RP only moving based on the "inferred" target of its human partner, without independent information about the target's trajectory?

In either case, how do the authors account for the stabilizing effect of the damping in the haptic connector to the RP on the smoothness of the hand trajectory? In other words, what would happen if the RP did not assist the human (position gain set to zero and q-dot_r set to zero with the same velocity gain in Eq. 2)? Might this also reduce the tracking error compared to the solo condition, where, from my understanding, no haptic torques are being applied to the participant's wrist? Could the action of the RP in fact be more through a mechanism of impedance matching rather than active assistance and haptic communication per se?

I find it also interesting that adding noise can elicit the perception of delay. Is this conditioned by the fact that the subjects were told that they would interact with a robot or a human? Was it stated or implied that the robot or human would be acting toward the same goal?

Aside from this methodological question, I wonder if the author's could consider other studies of haptic perception in face of delays (e.g. Pressman, A., Karniel, A., & Mussa-Ivaldi, F. A. (2006, February). Perception of delayed stiffness. In The First IEEE/RAS-EMBS International Conference on Biomedical Robotics and Biomechatronics, 2006. BioRob 2006. (pp. 905-910). IEEE.) And in view of the sharp change in predicted behavior for larger delays (Fig. 6D), I wonder if the author's have considered the so-called binding problem in face of delays? Might the subjects consider small delays as noise when the delay is small and then shift to a compensation as delay strategy when the delay is large, because, hypothetically, with the larger delay the haptic interaction is perceived as a delayed interaction from a separate entity (the RP) rather than a noise perturbation to a single physical interaction?

Minor remarks

Abstract: Perhaps "Physically connected humans ..." ?

Line 50: Are the experimental finding mentioned here findings from previous studies, or a forward reference to what will be reported here?

Line 78-79: That subjects more reliably detected delay when there was delay vs. when there was not is clear It would nevertheless be interesting to know to what extent subjects perceived delay when there was none, i.e. did the responses to the item "It seemed like I felt haptic interaction with a delay." elicit a response that was different from "strongly disagree"?

Line 75-81: Could you clearly state which items are used in each analysis, and if you relied on only a single item in each case, or if "perceived noise" (item 6) was somehow conditioned by responses to other items, such as "interaction with an agent" or not?

Line 85: How is this result "different to the results in the delay group". The sentence confused me because the delay group also perceived delay even for the smallest delay. I suggest: "Participants in the noise group clearly identified the presence of noise (χ2(5) = 66.411, p < 0.0001) as shown in Fig. 3D. In contrast to the perceived noise in the delay group, noise-group participants perceived the presence of noise from the smallest applied noise level (p < 0.01 for all pairwise comparisons between 0 mNm and 8-184 mNm noise conditions)."

Line 90: "none of the conditions were found to be clear different." From each other? Or from the solo or zero-delay conditions? A post hoc test of all pairwise comparisons would be less powerful then a set of planned comparisons between each of the non-zero delays and the zero-delay condition.

Fig. 3: I quibble with the axis titles "perceived as ..." and wonder if the delay is orthogonally independent from the noise. Unless the subject knew that there would be only noise or only delay, then the answers to the questionnaire tested only for the perceived presence of noise ("perceived noise") or the perceived presence of delay ("perceived delay"). Whether they perceived delay as noise is a matter of interpretation. Did any subjects perceive delay only as increased noise? The argument that delay is distinguishable from noise also depends on the assumption that no noise is actually added when adding delay. Does introducing delay in fact add noise to the haptic signals received by the subject via the coupling? Could a robotic system take the place of the human and, knowing that the RP will introduce only delay, could the robot identify the delay and produce trajectories that are no noisier than the non-delay system?

Line 134: The statement is factually correct, but the "in contrast to" perhaps invites an apples-to-oranges comparison. There is no way, in my mind, to calibrate the potential effect of a given level of noise on the tracking performance to the potential effect of a given level of delay on tracking performance. There is no reason to suppose and equivalence between 184 nMn of noise with 540 ms of delay in terms of effect on the tracking error in the absence of co-contraction, so the fact that the co-contraction in the case of delay did not prevent an increase of tracking error is potentially misleading.

Fig 6: What is the meaning of the different grey lines? Are 6A and 6B simply a different representation of the same results reported in 4A, but with connections between results for individual participants? What is the meaning of the dashed lines?

Why is 6B present, since in the text it is said "To evaluate this, we simulated the experimental scenario (with the same number of trials and blocks) of a participant being connected (by a virtual spring) to a partner with delayed haptic feedback."? Are the authors also simulating the compensation as noise strategy for subjects who received noise perturbations? If so, perhaps an explicit comparison of simulation results in 6E to 6B would also be warranted in the text.

What it the meaning of the grey lines for the simulation results (6C-6F)? Are these individual simulations tailored to each participant? The way that this was done is not clear to me. And if this is true, does 6C-F represent simulations for both the delay group and the noise group? Or just the delay group?

Lines 184-185: Since the RP is reacting to the haptic information from the human, I wonder if it is the same thing to delay the torque produced by the virtual spring vs. delaying the RP's "own target" estimate.

Lines 204-210: Have the authors confused 6D and 6F in the text?

Questionnaire:

What is the difference between items 3 and 7? i.e. what is the difference between "haptic forces" and "haptic feedback"? Were there differences in responses to these two items?

Were the answers to the questions internally consistent? For instance, if a subject answered non-zero for 8 or 9, did they also answer non-zero for 7 and zero for 12?

Reviewer #2: The paper generally addresses an interesting and important problem – haptic collaboration over delayed channels. This problem has many practical implications in today’s world and also may shed light on an interesting and unsolved problem of coping with internal delays (however this implication was not mentioned by the authors). The authors nicely extend their previous studies of collaboration between humans and robotic agents in a task of tracking a target, and in the current study they investigate how this interaction is affected by adding noise or by adding a delay. While the only new experiment is the effect of noise, I do appreciate the analysis of the old and the new study together to make a point and the previous work is correctly acknowledged in the paper. Overall the work is of good quality. However, I have the following major reservations about this work:

(1) The authors completely ignore the existing literature about how delays affect movement control and perception. While the particular task of haptic collaboration over a delayed channel was not extensively studied, the effect of delay was studied in several contexts that are related to the task. Except from the failure to put the study in the correct context in terms of prior literature, these prior studies could yield several other possible models for coping with delay except from modeling it as noise. Delay in force feedback was previously shown to change the perception of the mechanical properties of environment (stiffness, mass). Delay in force field adaptation was proposed to be compensated as a combination of current and delayed movement signals, or as a state-based approximation – position, velocity, and acceleration). Delay in visual feedback was proposed to be compensated as a gain, as a mechanical system, or as an altered inertia). All these could result in possible models that can propose explanations for the inaccurate coping with delay beyond noise. While the results of this paper nicely show that delay is not compensated as noise, suggesting that some type of representation or approximation of delay exists, I wander does it make sense to assume such complete lack of representation as an alternative to accurate prediction of time when evidence in favor of alternative ideas exists that are not noise but also not delay representation?

Here are a few of the references that are relevant. Note that even though quite a few of these papers are from my group, (not all though), by all means i do not request the authors to cite this entire list. But i would like the author to be aware of these studies and decide how to proceed with this knowledge:

Pressman et al., 2007, Perception of delayed stiffness

Nisky et al., 2008, A regression and boundary-crossing based model for the perception of delayed stiffness

Nisky et al., 2010 A regression and boundary-crossing based model for the perception of delayed stiffness

DiLuca et al., 2011 Effects of visual–haptic asynchronies and loading–unloading movements on compliance perception

Leib et al., 2015 The effect of force feedback delay on stiffness perception and grip force modulation during tool-mediated interaction with elastic force fields

DiLuca Rhodes 2016 Optimal Perceived Timing: Integrating Sensory Information with Dynamically Updated Expectations

Leib et al., 2017 The mechanical representation of temporal delays

Avraham et al., 2017 “Representing Delayed Force Feedback as a Combination of Current and Delayed States

Avraham et al., 2017 State-based delay representation and its transfer from a game of pong to reaching and tracking

Farschian et al., 2018 Energy exchanges at contact events guide sensorimotor integration across intermodal delays

Avraham et al., 2019 Effects of Visuomotor Detlays on the Control of Movement and on Perceptual Localization in the Presence and Absence of Visual Targets

Van Polanen et al., 2019 Visual delay affects force scaling and weight perception during object lifting in virtual reality

(2) It is not entirely clear to me how the computational model supports the experimental findings beyond the conclusion that can be reached from the error and co-contraction results? The predictions of the model were consistent generally with the experimental results, but it is difficult to tell if the chosen model indeed explains what happens in the delay case. Indeed the authors are right in their choice of the title stating that the main finding is that delay and noise are not compensated similarly. But this conclusion does not require the computational model. The alternative models I mentioned in the previous point could possibly propose a more compelling explanation.

In addition, I think the following points could further improve the paper, regardless to my previous comments:

1) It would help the interpretations a lot if the authors would explain exactly what are their prediction with respect to each of the computed metrics, and mentioned the specific results in the narrative rather than just the statistical results. For me the result that was most difficult to understand due to the lack of predictions and narrative is the results of the delay cross-correlation – did they expect the delay to change? To which value? What the exact value of the delay teaches us? The authors state in the discussion that: “The reduced lag then suggests that this information not only improves the quality of their estimation but also aids them to adapt quicker then what they normally would” but it is not clear to me why. The other results are a bit more intuitive, but still the way they are presented could be improved – I would like to be able to get an idea about the exact result from the text, and then go to the figure to make sure I agree with the authors based on what I see.

2) Similarly, the exact interpretation of the simulations are not entirely clear. For example, I could not tell based on figure 6 whether a real difference exists between the predictions of the no-compensation and the compensation as delay, which is confusing and makes the message of the paper less clear.

3) The illustration of the different models in Figure 1 is also not clear – why at some point the compensation as delay diverges from the original trajectory. In general, in addition to the general illustration becoming more clear, I would like to see simulated trajectories and experimental trajectories to be able to compare the predictions of the model to the data.

4) Minor: they call the connection between the partners an elastic band but I believe a viscoelastic band is more accurate.

To conclude, I like the direction where the authors are going, but I think that the current work does not address sufficiently the models from the literature, and does not provide enough information about the simulation and its interpretation to merit the paper publication in its current form.

**Have the authors made all data and (if applicable) computational code underlying the findings in their manuscript fully available?**

Reviewer #1: Yes

Reviewer #2: **No: **

PLOS authors have the option to publish the peer review history of their article (what does this mean?). If published, this will include your full peer review and any attached files.

Reviewer #1: No

Reviewer #2: **Yes: **Ilana Nisky
---

## [Decision Letter · Decision Letter 1]

18 Oct 2024

Dear Dr Eden,

We are pleased to inform you that your manuscript 'During haptic communication, the central nervous system compensates distinctly for delay and noise' has been provisionally accepted for publication in PLOS Computational Biology.

Best regards,

Adrian M Haith

Academic Editor

PLOS Computational Biology

Andrea E. Martin

Section Editor

PLOS Computational Biology

Reviewer's Responses to Questions

**Comments to the Authors:**

Reviewer #1: I am satisfied with the revised manuscript and I appreciate the careful attention that the authors have given to addressing my questions and concerns.

**Have the authors made all data and (if applicable) computational code underlying the findings in their manuscript fully available?**

Reviewer #1: Yes

PLOS authors have the option to publish the peer review history of their article (what does this mean?). If published, this will include your full peer review and any attached files.

Reviewer #1: No

---

## [Editor Report · Acceptance letter]

29 Oct 2024

PCOMPBIOL-D-24-00534R1 

During haptic communication, the central nervous system compensates distinctly for delay and noise

Dear Dr Eden,

I am pleased to inform you that your manuscript has been formally accepted for publication in PLOS Computational Biology. Your manuscript is now with our production department and you will be notified of the publication date in due course.

With kind regards,

Zsofia Freund
